# Engineering the auxin-inducible degron system for tunable in vivo control of organismal physiology

Jeremy Vicencio [1], Daisuke Chihara[1], Matthias Eder[1], Lucia Sedlackova [1], Julie Ahringer[2] & Nicholas Stroustrup [1,3] ✉

The auxin-inducible degron (AID) system is designed for the rapid and near-complete degradation of a specific target protein in vivo. However, to understand the dynamics of complex physiological networks, researchers often need methods that produce graded, quantitative changes in degradation rates for multiple proteins simultaneously. Here, we develop the AID system for in vivo, quantitative control over the abundance of multiple proteins simultaneously. First, by measuring and modeling the on- and off-target activities of different AID system variants in *Caenorhabditis elegans*, we characterize a variant of the E3 ubiquitin ligase subunit TIR1, which provides improved degradation activity compared to the original AID and AID2 systems. Then, we develop a TIR1 expression construct that enables simultaneous pan-somatic and germline protein degradation. Finally, we expand the AID toolkit to allow independent, simultaneous degradation of two distinct tissue-specific proteins. Together, these technologies enable new in vivo approaches for studying quantitative cellular biology and organismal dynamics.

In the field of molecular genetics, the causal role of a gene in a phenotype is established by experimentally perturbing its function, often using interventions that produce strong protein knockdown or knockouts. When biological systems exhibit complex quantitative dynamics, strong knockdowns may be less informative than more subtle, quantitative perturbations[1–3]. A variety of methods exist to provide finer spatiotemporal control of gene expression, including recombination with the Cre-lox[4] and FLP-FRT[5] systems, promoter manipulation using CRISPR-Cas genome editing[6,7], dose-dependent RNA interference (RNAi)[8], or protein degradation using small-molecule proteolysis targeting chimeras[9]. The auxin-inducible degradation (AID) system[10] has many practical advantages over other methods, including rapid and reversible degradation. In particular, the involvement of an easily-deliverable, small-molecule activating compound makes the AID system a particularly promising technology for quantitative control over protein abundance in vivo.

The AID system evolved in plants as a mechanism through which the hormone auxin regulates proteins required for growth and development[11]. Adapted first into yeast and a wide range of eukaryotic cells[10], then *Caenorhabditis elegans*[12], *Drosophila*[13], and mice[14], researchers have found that two components of the AID system are sufficient to support targeted protein degradation in animals: first, the F-box transport inhibitor response 1 (TIR1) protein which interacts with the Skp1-Cul1-F-box (SCF) E3 ligase complex to polyubiquitinate target proteins, and second, a short amino acid tag (called a "degron") that is fused to target proteins and acts as a TIR1 substrate[15] (Fig. 1a). The TIR1 induces ubiquitination of degron-tagged proteins only in the presence of the small-molecule indole-3-acetic acid (IAA), providing a means for small-molecule-induced proteasomal degradation[16]. The high degree of conservation of Skp1 and Cul1 among eukaryotes allows transgenic TIR1 to interact with the Skp1 and Cul1 proteins endogenous to the host species[17,18], removing the need for transgenic E3-Ubiquitin ligases.

[1]Centre for Genomic Regulation (CRG), The Barcelona Institute of Science and Technology, Barcelona, Spain. [2]The Gurdon Institute and Department of Genetics, University of Cambridge, Cambridge, UK. [3]Universitat Pompeu Fabra (UPF), Barcelona, Spain. ✉e-mail: nicholas.stroustrup@crg.eu

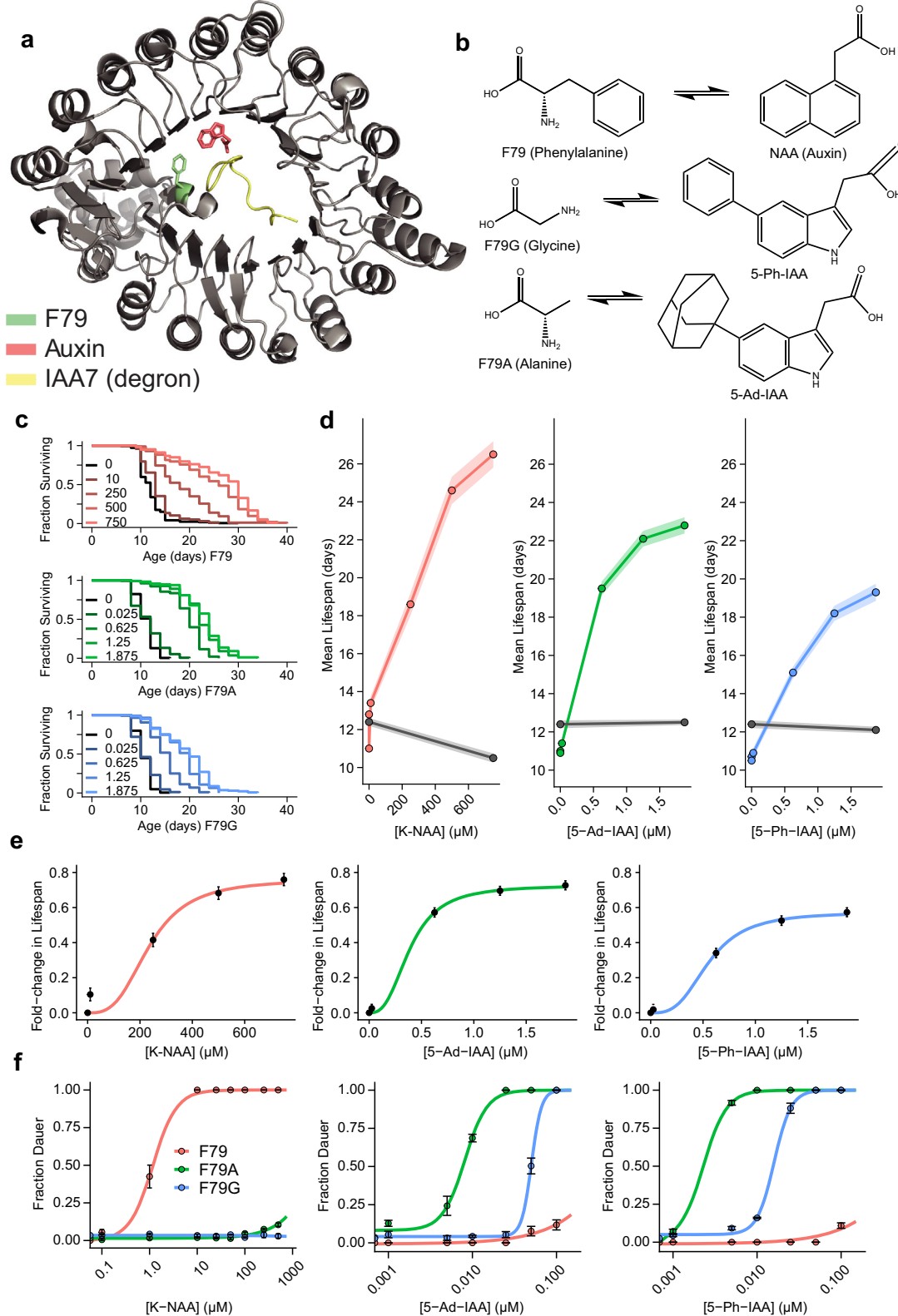

**Fig. 1 | The structure and quantitative activity of three TIR1 variants. a** The crystal structure of *At*TIR1 (adapted from Tan et al., 2007)[15]. **b** The three amino acids considered in the auxin-binding pocket of TIR1, along with their high-affinity ligands. **c** Kaplan–Meier survival curves showing the dose–response effect on lifespan of K-NAA with TIR1[F79] (red), 5-Ad-IAA with F79A (green), and 5-Ph-IAA with F79G (blue), in populations expressing DAF-2::AID and the respective TIR1 variant. The compound concentrations are in μM. **d** The mean (points) and ±1 standard error (shaded region) of the same populations (colors) compared to wild-type (black). **e** The fold-change in lifespan observed in the same populations, fit by Hill functions. **f** The same TIR1[F79] (red), F79A (green), and F79G (blue) strains, but this time, assaying for a constitutive dauer entry phenotype on K-NAA (left), 5-Ad-IAA (center), and 5-Ph-IAA (right) (*N* = 36,225 individuals). The data for (**c–e**) were derived from *N* = 1714 individuals from four independent experiments. Error bars represent ±1 SE (standard error).

The AID system has been shown to be capable of depleting most of the target protein between 15 min and 3 h, depending on the half-life of the target protein[10,19]. In addition, this depletion is rapidly reversible, with recovery time being inversely proportional to the concentration of auxin used[12]. In contrast, RNAi techniques involve a significant time delay between induction and depletion of the target[20], and are not necessarily reversible due to the relatively long half-life of small interfering RNAs once loaded into the RNA-induced silencing complex[21].

A few variants of the AID system have been developed to optimize its performance and enable new experimental techniques. In plants, botanists designed an orthogonal small-molecule–TIR1 pair using a "bump-and-hole" strategy[22] that alters the TIR1 ligand-binding pocket to recognize a chemically modified auxin derivative, providing a means for target protein degradation independent of the endogenous AID system. This system was subsequently adapted for use in animals, branded as the "AID version 2" (AID2)[23]. In this system, a single amino acid substitution, F79G, increases the affinity of *Arabidopsis thaliana* TIR1 (F74G in *Oryza sativa* TIR1) to the synthetic auxin analog 5-phenyl-indole-3-acetic acid (5-Ph-IAA) (Fig. 1b). TIR1[F79G] ubiquitination activity can be induced at 670-fold lower concentrations of 5-Ph-IAA compared to those required of IAA to activate the original TIR1[F79]. The AID2 system has been reported to exhibit faster depletion kinetics than the original system and lower "basal" auxin-independent ubiquitination activities[24,25].

A second TIR1 variant has been developed using a distinct mutation at the same TIR1[F79] residue: F79A (*Os*TIR1[F74A]) to increase TIR1 affinity to the synthetic auxin analog 5-adamantyl-IAA (5-Ad-IAA, Fig. 1b). This variant was named "super-sensitive AID (ssAID)" and can be activated by 5-Ad-IAA at 1000- to 10,000-fold lower concentrations compared to those required of IAA for TIR1[F79], depending on the cell type[26,27]. Unlike F79G, F79A has not yet been adapted for use in *C. elegans*.

The AID system is now widely used across eukaryotic systems but has seen particularly rapid adoption in *C. elegans* research. Since its introduction in *C. elegans*[12], the AID system has been progressively refined by adopting synthetic auxin analogs like NAA and K-NAA, which offer superior photostability and water solubility for rapid and reversible protein depletion[28,29]. Versatile protocols have been developed for long-term, acute, and tissue-specific depletion across various formats[30–32]. Subsequent innovations include orthogonal *At*TIR1[F79G]/5-Ph-IAA and 5-Ph-IAA-AM pairings that suppress ligand-independent degradation[25] and lower effective ligand doses in embryos[24], and minimal and multimerized degron motifs (mIAA7) that enhance degradation kinetics of challenging protein targets[33]. In addition, a toolbox comprising single-copy, tissue-specific, and pan-somatic TIR1-expressing strains, as well as modular CRISPR/Cas9 repair template toolkits that facilitate AID tagging are available[34]. Moreover, several *C. elegans* studies have independently reported the ligand-independent degradation of AID-tagged proteins, also referred to as basal degradation[25,28,35,36], and that the adoption of the AID2 system results in lower basal degradation compared to the original AID system[24,25].

Here, we develop the AID system to obtain quantitative control over protein abundance in vivo by systematically measuring the kinetics and dose–response curves of multiple degron–promoter–TIR1 combinations. We then use our data to guide the design of new TIR1 variants with enhanced degradation efficiency and tunable activity. We further enhance spatiotemporal control by engineering orthogonal TIR1 constructs for simultaneous, independent degradation of different targets in distinct tissues and by re-engineering TIR1 for robust, pan-organismal protein depletion in *C. elegans*.

## Results

### TIR1 variants differ in their dose-response relationships

Three versions of the AID system currently exist, based on the identity of the amino acid at position 79, located on the side wall of the auxin-binding pocket of the *Arabidopsis thaliana* TIR1 (*At*TIR1) protein. In wild-type TIR1, there is a phenylalanine residue at position 79 (F79, Fig. 1b). Orthogonal auxin–TIR1 pairs have been developed based on a "bump-and-hole" approach where the side wall of TIR1 is carved by substituting F79 with an aliphatic residue while reciprocally introducing a bump to the indole ring of IAA to fill the cavity. In the AID2 system, F79 is substituted with glycine (F79G) and is paired with its high-affinity ligand 5-Ph-IAA. In contrast, F79 is substituted with alanine (F79A) in the ssAID system and is paired with its high-affinity ligand 5-Ad-IAA (Fig. 1b). To allow rapid conversion among these TIR1 variants, we designed a CRISPR-Cas9 strategy to edit the F79 locus of TIR1 transgenes in vivo (Supplementary Fig. 1) to obtain three identical *C. elegans* strains differing only in a single amino acid residue. We obtained editing efficiencies of 51.4% for F79 to F79A conversion and 45.9% for F79 to F79G conversion from 37 genotyped animals each, with some homozygous conversions occurring in the first filial generation ($F_1$). We conclude that our method enables rapid conversion between different engineered TIR1 alleles, enabling users to select the appropriate one for their purpose.

To compare the quantitative performance of these TIR1 variants in a physiologically relevant context, we chose the *daf-2* insulin/IGF receptor as a degradation target. *daf-2* is a highly pleiotropic gene that exerts a quantitative effect on multiple phenotypes. Mutations in the insulin receptor can extend lifespan up to 300%[37], a phenotype recently recapitulated using the AID system[38]. Mutations in *daf-2* are also known to influence a complex organismal decision made by *C. elegans* during development, in which a variety of sensory cues are integrated to decide whether or not to enter a spore-like diapause state[39]. Therefore, choosing *daf-2* as a target for engineering the AID system provides us with multiple quantitative phenotypes to measure AID degradation rates.

First, we focused on the lifespan phenotype, which allows us to quantify the integrated effect of TIR1 activity over several weeks. In an experiment involving 7400 individuals, we observed a dose-dependent increase in lifespan extension in all three TIR1 variants, identifying a range in which the kinetics of protein depletion is rate-limited by auxin ligand concentration (Fig. 1c). At the highest concentrations of ligands tested (750 μM K-NAA, 1.875 μM 5-Ph-IAA and 5-Ad-IAA), we observed that the original F79 variant was able to produce the highest mean lifespan (26.5 days), followed by F79A (22.8 days), and finally, F79G (19.3 days, Fig. 1d). In the absence of any TIR1 variant, 750 μM K-NAA produced a 15.5% decrease in lifespan ($p = 10^{-9}$) whereas 1.875 μM 5-Ph-IAA and 5-Ad-IAA did not affect lifespan ($p = 0.02$, $p = 0.30$, respectively; Supplementary Fig. 2a). Notably, the presence of the F79A and F79G TIR1 variants in a DAF-2::AID background appeared to have a small lifespan shortening effect (11% and 14%, respectively, $p < 10^{-6}$), suggesting a minor "off-target" lifespan-shortening effect of the two TIR1 variants (Supplementary Fig. 2b).

To better understand the kinetics of protein depletion, we fit Hill functions to our dose–response data. This allowed us to estimate the maximum lifespan extension at saturating ligand concentrations for each TIR1 variant: 219% in F79, 203% in F79A, and 184% in F79G (Fig. 1e; dotted lines). These results again suggest that the F79 variant produces the greatest DAF-2::AID degradation among the three variants.

To compare the cross-reactivity of the three ligands across the three TIR1 variants, we turned to the dauer assay, which allows us to quantify DAF-2 degradation during development instead of adulthood. Between the first and second larval molts, DAF-2 receptor activity determines the probability that an individual will enter diapause, a morphologically distinct state we can visually identify and use as a quantitative readout of TIR1 activity. Characterizing 36,225 individuals, we find that much lower concentrations of ligand are required to shift animals from 0 to 100% dauer entry compared to the doses required to extend lifespan (Fig. 1f). Again, using Hill fits, we estimated the ligand concentrations required to produce 50% dauer entry ($EC_{50}$) and

compared these concentrations across TIR1 variants. For TIR1[F79], the $EC_{50}$ of this variant's canonical ligand K-NAA was 1.2 μM. The $EC_{50}$ was never reached for the other two compounds. For TIR1[F79A], the $EC_{50}$s were 8.1 nM for the canonical ligand 5-Ad-IAA, 2.3 nM for 5-Ph-IAA, and was never reached for K-NAA. Finally, for TIR1[F79G], the $EC_{50}$s were 15.9 nM for the canonical ligand 5-Ph-IAA, 50.5 nM for 5-Ad-IAA, and the $EC_{50}$ was never reached for K-NAA. From these data, we conclude that TIR1[F79] shows little cross-reactivity with 5-Ad-IAA and 5-Ph-IAA at their typical concentrations, that TIR1[F79A] and TIR1[F79G] cross-react with both 5-Ph-IAA and 5-Ad-IAA, respectively, and that TIR1[F79A] shows slightly more cross-reactivity with K-NAA than TIR1[F79G].

To measure the stability of all three compounds and quantify their ability to activate TIR1 consistently during long-term assays, we performed a diagnostic experiment in which DAF-2::AID; TIR1 populations sensitive to each compound were shifted onto aged plates. Plates were poured and seeded with *Escherichia coli* OP50 as the food source, then stored at room temperature for up to 28 days. At 28 days, we found that 99.7, 92.4, and 93.6% of individuals entered dauer in response to 5-Ad-IAA, 5-Ph-IAA, and K-NAA respectively (Supplementary Fig. 3a). Aging K-NAA plates in the absence of *E. coli*, and seeding them one day before the assay, increased the dauer entry rate to 99.3%, highlighting the dominant role of bacteria in plate aging (Supplementary Fig. 3b). Nevertheless, despite the sensitivity of our dauer assay, the small changes in compound activity observed even after a full month of aging demonstrate a high degree of compound stability suitable for quantitative, long-term experiments.

## TIR1 enzymatic activity is primarily determined by its abundance, with variant-specific limitations

We then applied an interventional approach to identify the biochemical mechanisms determining the AID system performance. We experimentally modulated two aspects of the system: first, we introduced a 3× repetition of the degron sequence on the *daf-2* receptor (Fig. 2a). Our version of the 3× degron tag uses three copies of AID*, in contrast to the 3× mIAA7 tag previously found to improve auxin-dependent degradation at the expense of increased basal degradation[33]. Second, we altered the abundance of the TIR1 protein by replacing the somatic *eft-3* promoter with an *eif-3.B* promoter that results in approximately 2.0-fold lower somatic expression (Fig. 2b and Supplementary Fig. 2c). In addition, the *eif-3.Bp*::TIR1 line results in measurable germline expression (Fig. 2c and Supplementary Fig. 2c), creating the potential for pan-organismal degron knockdown. Having made these constructs, we then explored several permutations of the expanded AID system: pairing both 1×AID and 3×AID degron tags with the F79, F79A, and F79G TIR1 variants.

We then quantified the effect of these modifications to the AID system using lifespan across a population of 7400 individuals as a readout of DAF-2 degradation rates. For the wild-type TIR1[F79] construct, we find that with the standard 1×AID tag, reducing TIR1 using *eif-3.B*p reduces the maximum lifespan extension obtainable (Fig. 2d). However, by adding a 3×AID tag, the original lifespan extension can be restored. In contrast, adding a 3×AID tag to the highly expressed *eft-3p*::TIR1 line produced only small additional lifespan extensions. These data reveal important biochemical determinants of the AID system: at saturating ligand concentrations, *eft-3p*::TIR1 is sufficiently abundant to ubiquitinate DAF-2::AID to the maximum extent, thereby maximizing degradation rates with 1 degron repeat. However, after lowering TIR1 abundance using *eif-3.Bp*::TIR1, even at saturating K-NAA concentrations, the available TIR1 pool is not sufficient to maximally ubiquitinate DAF-2::AID. Then, our data demonstrate that adding 3×AID repeats to DAF-2::AID increases degradation rates, allowing each TIR1 molecule to drive greater degradation, presumably through cooperative interactions among the additional ubiquitination sites available on the 3×AID.

In contrast, pairing the 3×AID tag with the TIR1[F79A] mutation substantially increased the lifespan extension produced even when the TIR1 protein was expressed at high abundance under the *eft-3* promoter (Fig. 2e). This suggests that the rate-limiting factors for TIR1[F79A]-mediated DAF-2::AID degradation are different from those for wild-type TIR1[F79]. The lower lifespan extension of TIR1[F79A] compared to TIR1[F79] in the 1×AID background highlights a lower intrinsic ubiquitination rate for each molecule of TIR1[F79A] compared to TIR1[F79], since the former fails to maximize ubiquitination rates despite sufficient TIR1 abundance. However, the intrinsic lower activity of F79A is compensated for by pairing it with a 3× degron, as demonstrated by the largest lifespan extension produced by any combination of TIR1 variants and degron tags (Fig. 2e). In contrast, pairing the 3×AID tag with TIR1[F79G] produced no effect on lifespan compared to 1×AID (Fig. 2f). These data suggest that the intrinsic activity of TIR1[F79G] is not only lower than TIR1[F79] and TIR1[F79A], but also that this defect cannot be compensated for by adding additional ubiquitination sites.

Finally, we compared all combinations of TIR1 and degron tags using the dauer assay in a population of 78,750 individuals. For all strains, we again obtained 100% dauer entry rates (Fig. 2g–i). In concordance with our lifespan data, lowering the expression of TIR1[F79] using the *eif-3.B* promoter decreased the apparent degradation rate of DAF-2::AID, increasing the $EC_{50}$ 150-fold from 1.2 μM to 173 μM (Fig. 2g), once again highlighting the crucial dependence of AID system activity on TIR1 expression levels. Adding a 3× degron recovered some of this activity, lowering the $EC_{50}$ by about 40%. This lowering of the $EC_{50}$ was observed both in *eft-3p*::TIR1 and *eif-3.Bp*::TIR1 (Fig. 2g), suggesting that during development, *eft-3p*::TIR1 is not expressed at sufficiently high concentrations to produce maximal ubiquitination rates with 1×AID.

Adding 3× degrons to the TIR1[F79A] and TIR1[F79G] variants again produced very large decreases in the $EC_{50}$ of their respective ligands—much larger than the effect of 3× degron for the TIR1[F79] strain (Fig. 2h, i). These data support our model that TIR1[F79A] and TIR1[F79G] produce lower per-molecule ubiquitination rates of DAF-2, deficits that can be compensated for by increasing the availability of ubiquitination sites. In the lifespan assay, we had found that adding a 3×AID tag produced little to no effect on TIR1[F79G] activity (Fig. 2f), but in the dauer assay (Fig. 2i), the effects we see from adding the 3×AID tag suggest that adding additional ubiquitination sites can improve TIR1[F79G] activity when TIR1[F79G] is expressed at lower than saturating abundances—a situation we infer is the case during development but not during adulthood.

In conclusion, our data reveal important determinants of AID activity, most notably the requirement that TIR1 be expressed at high abundance relative to its degradation targets to ensure that TIR1 abundance is not the limiting factor for degradation rates. Regardless of other benefits of the TIR1[F79A] and TIR1[F79G] variants, we find that these two proteins exhibit substantially lower overall TIR1 ubiquitination activity per molecule, to the extent that TIR1 abundance is limiting even at very high expression levels. For TIR1[F79A] but not TIR1[F79G], this deficit can be ameliorated by adding a 3×AID tag to the degradation target.

## All three TIR1 variants produce large, "on-target" effects on gene-expression via DAF-2 degradation

We then sought to apply a third quantitative assay to systematically evaluate the "on-" and "off-target" activities of the AID system across multiple physiological aspects beyond dauer entry and lifespan. We reasoned that because *daf-2* mutations produce strong, well-characterized effects on organismal gene expression[40], we could use RNA-sequencing to identify both on-target effects of the AID system mediated by DAF-2 degradation as well as off-target effects involving other genes. Thus, we collected mRNA from the eight TIR1-expressing

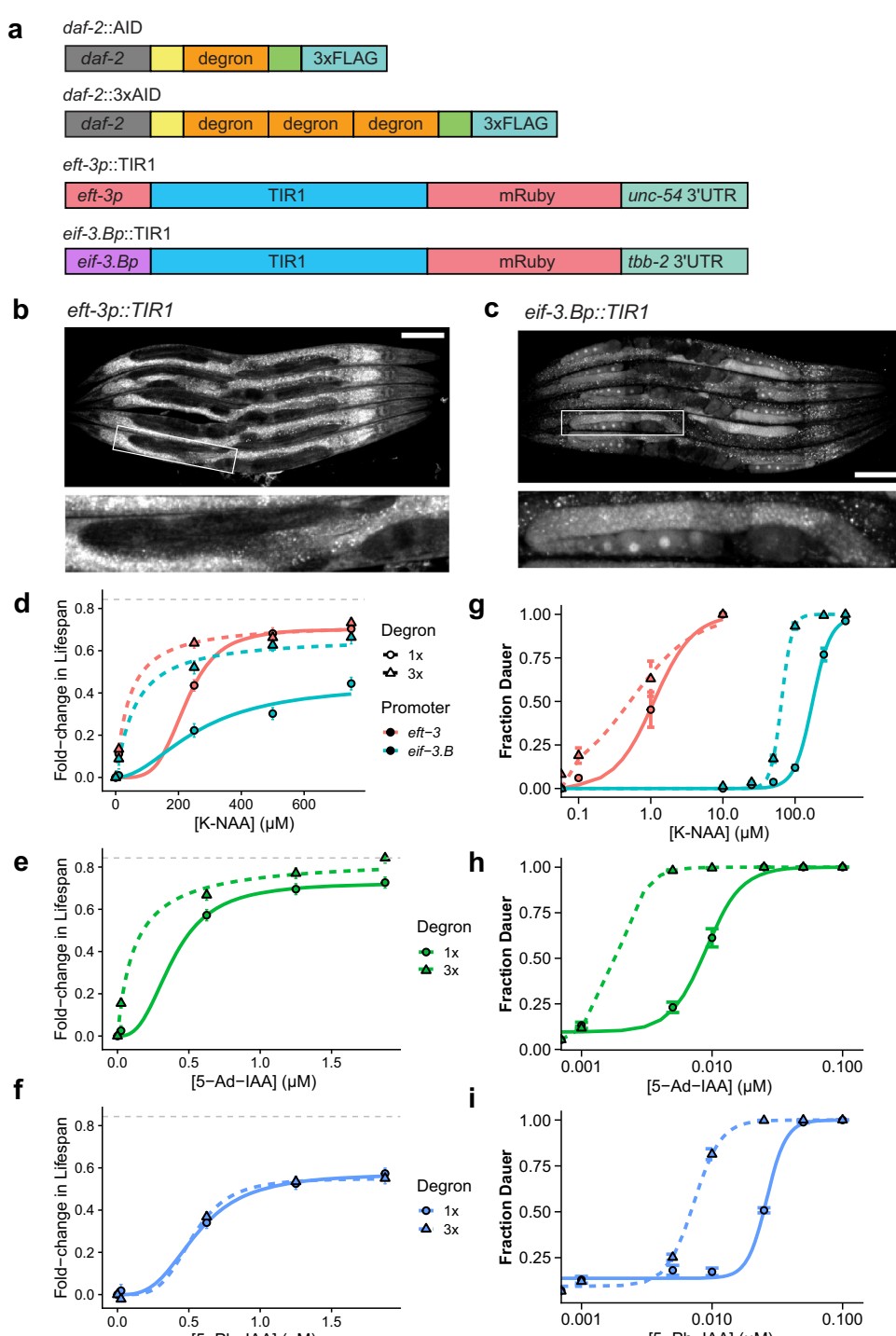

**Fig. 2 | Engineering quantitative control of the auxin-inducible degradation (AID) system. a** Fine-tuning the AID system by modulating the number of degron repeats on the AID tag (top) or modulating TIR1 expression levels (bottom) using the *eft-3* promoter (high somatic expression, no germline expression) or the *eif-3.B* promoter (intermediate somatic expression, high germline expression). **b** Organismal expression of *eft-3p*::TIR1::mRuby, with germline (inset) expression absent. **c** Organismal expression of *eif-3.Bp*::TIR1::mRuby, with the germline (inset) expression present (scale bars = 100 μM). **d** Lifespan dose-response to K-NAA in populations expressing TIR1[F79] under the *eif-3.B* promoter (blue), *eft-3* promoter

(red), combined with DAF-2::AID (solid lines; circles) or DAF-2::3×AID (dotted lines; triangles) (*N* = 4457 individuals). **e** The dose response of 5-Ad-IAA on populations expressing *eft-3p*::TIR1[F79A] combined with DAF-2:AID (solid lines; circles) or DAF-2::3×AID (dotted lines; triangles) (*N* = 1481 individuals). **f** The dose response of 5-Ph-IAA on populations expressing *eft-3p*::TIR1[F79G] combined with DAF-2::AID (solid lines; circles) or DAF-2::3×AID (dotted lines; triangles) (*N* = 1435 individuals). **g**–**i** The dose response of each of the compounds and TIR1 variations in (**d**–**f**), but this time characterizing the probability of dauer entry (*N* = 78,750 individuals). The error bars in (**d**–**i**) represent ±1 SE.

strains considered so far: *eft-3p*::TIR1[F79], *eif-3.Bp*::TIR1[F79], *eft-3p*::TIR1[F79A], and *eft-3p*::TIR1[F79G] each combined with either DAF-2::AID or DAF-2::3×AID—with and without 24 h of exposure to the highest concentration of their respective ligands, K-NAA (750 μM) and 5-Ph-IAA or 5-Ad-IAA (1.875 μM).

Principal component analysis (PCA) revealed strong effects of the activated AID system on gene expression (Fig. 3a). The first principal component (PC), explaining 52% of all inter-sample variation (Supplementary Fig. 4a), reveals the effect of ligand-mediated induction on strains expected to degrade DAF-2::AID (high PC1 values). We find that the gene expression effects quantified along PC1 capture an aspect of gene expression on day 2 that is highly correlated with each strain's remaining lifespan (Pearson correlation $\rho$ *(rho)* = 0.971; Fig. 3b; Supplementary Fig. 4a), demonstrating that changes in the gene expression state along PC1 are associated with the "on-target" lifespan-extending effects of DAF-2 degradation, even though its effects on lifespan are revealed several weeks after measuring its influence on gene regulation on day 1. This high degree of correlation between gene regulation and lifespan is observed even when strains nonresponsive to activating compounds are excluded from analysis (Supplementary Fig. 4b, c).

We observe that DAF-2::AID degradation via 5-Ad-IAA-activated TIR1[F79A] produces similar effects as K-NAA-activated, TIR1[F79]-mediated degradation ($\rho$ = 0.54, Fig. 3c), though F79A produced these effects at a smaller magnitude: only 63% on average compared to F79 (robust nonlinear regression, 95%CI: 61–65%). DAF-2::AID degradation mediated by 5-Ph-IAA-activated TIR1[F79G] produced even smaller effects: only 45% (43–47%) as large as in F79 ($\rho$ = 0.34, Fig. 3d), consistent with the smaller magnitude of effects we saw in lifespan assays using TIR1[F79G]. Overall, TIR1[F79A] produced 40% larger effects on gene expression (37–42%) relative to TIR1[F79G] ($\rho$ = 0.53, Fig. 3e,f). Not surprisingly, the common targets differentially regulated by DAF-2 knockdown across all three TIR1 variants (Supplementary Data 3) were highly enriched for genes differentially regulated by DAF-2 (qval <$10^{-36}$) as measured by 34 independent experimental data sets (WormExp) and DAF-16 targets (qval <$10^{-3}$) across 23 independent data sets (WormExp, Supplementary Data 4).

DAF-2 knockdown mediated by *eif-3.Bp*::TIR1, expressed in a pan-organismal expression pattern but at lower levels compared to *eft-3p*::TIR1, produced very similar effects as *eft-3p*::TIR1 ($\rho$ = 0.70, Fig. 3g). The presence of additional degron repeats in the DAF-2::3×AID population increased the magnitude of gene expression changes by 18% compared to DAF-2::1xAID ($\rho$ = 0.79, Fig. 3h), again consistent with the relative magnitude of effects observed in our lifespan and dauer phenotypic data.

## The source of undesirable activities of the AID system, including basal activity

Our phenotypic and transcriptomic data highlight important, previously unreported differences in the performance of AID2 TIR1[F79G] compared with TIR1[F79]. In any TIR1 system, a major concern has been the "basal activity" observed in the absence of an activating compound. To better understand the mechanistic basis of AID system activities, we applied our transcriptomic approach to dissect the individual and combined effects of each AID system component—TIR1 transgenes, AID degron tags, and auxin-analog compounds.

### K-NAA, 5-Ph-IAA, and 5-Ad-IAA interact epistatically with TIR1. To understand the direct (i.e., TIR1-independent) effects of auxin analogs on physiology, we measured the effect of each compound on wild-type populations. We observe a strong effect of K-NAA on gene expression (Fig. 4a left, d), enriched for genes in the E3 ubiquitin proteasome family. In contrast, 5-Ph-IAA and 5-Ad-IAA both produce comparatively smaller effects on gene expression in wild-type animals (Fig. 4b, c left, e, f; Supplementary Fig. 5a, b and Supplementary Table 1), consistent

with their lower concentrations compared to K-NAA, confirming a clear advantage of these AID variants relative to TIR1[F79]. We find that the small remaining effects of 5-Ph-IAA and 5-Ad-IAA are nearly identical ($\rho$ = 0.93, Supplementary Fig. 5c).

**The AID tag modulates DAF-2 activity.** We find that adding a degron tag to DAF-2 modulates the expression of about 150 genes (Supplementary Fig. 5d–f), notably downregulating sperm-related genes such as *msd-3*, *msp-40*, and *msp-76* (Supplementary Table 2 and Supplementary Data 2, 4). Changes in most of these genes are not observed in populations also expressing the TIR1 variants F79, F79G, or F79A ($\rho$ = 0.1, 0.16, 0.12) (Fig. 4a–c center left), suggesting that the basal activity of the AID system to degrade DAF-2 counteracts the influence of the AID tag itself.

**TIR1[F79G] and TIR1[F79A] alter gene expression more than TIR1[F79].** The presence of the TIR1[F79] transgene alone had comparatively few effects on gene regulation (Fig. 4a center right, d), but in contrast, TIR1[F79G] produced larger and more numerous effects (Fig. 4b center right, e) in the absence of compound or degron tags. TIR1[F79A] fell somewhere in between the other two variants, producing more effects than F79 but less than F79G (Fig. 4c center right, f and Supplementary Fig. 5g–j). This degron-independent basal activity is an unrecognized disadvantage of the AID2 and ssAID systems: F79G, and to a lesser extent, F79A, appear to activate AID degron-independent activities that are not present in the original F79 progenitor (Supplementary Fig. 5k).

**TIR1[F79G] and TIR1[F79A] have higher basal activity than TIR1[F79].** Basal activity in the absence of activating compound is a major limitation of the AID system[28,35,36] in TIR1; DAF-2::AID animals. Using transcriptomics, we observe a higher basal activity of the F79G and F79A TIR1 variants, with 115 and 121 genes altered in TIR1[F79G] and TIR1[F79A], respectively, compared to 34 genes significantly altered by TIR1[F79] (Fig. 4d–f and Supplementary Table 3). Such effects on gene expression are consistent with the much higher rates of dauer entry in the absence of activating ligand in TIR1[F79G] and TIR1[F79A] strains (Supplementary Fig. 3c). Our results suggest that the lower off-target "basal" activity previously reported for the AID2 system may not be general across all applications[23–25], due to inherent differences in activity introduced by the amino acid substitutions.

Together, our systematic characterization of the AID system provides a better understanding of its strengths and limitations. Rather than being an unequivocal improvement over the original TIR1[F79] enzyme, the "AID2" TIR1[F79G] eliminates the unwanted effects of K-NAA at the cost of introducing much higher basal activities of the TIR1 itself, which alters physiology both through degron-dependent and degron-independent mechanisms. In addition to higher basal activities, TIR1[F79G] also exhibits lower "on-target" activities, as we find that the magnitude of DAF-2 knockdown is lower when measured using both gene expression and lifespan assays. The main advantage of the AID2 system, therefore, does not come from an improved enzymatic activity per se, but rather, from the lower TIR1-independent activity of 5-Ph-IAA relative to K-NAA. Meanwhile, the "ssAID" system we introduce here to *C. elegans* combines 5-Ad-IAA, a compound with similarly low AID-independent activity as 5-Ph-AA, with a TIR1[F79A] variant that exhibits similar basal "leaky" activity but superior "on-target" activities compared to the TIR1[F79G] of AID2. By decomposing the effects of the AID system into contributions from its individual components, we obtain a better understanding of the mechanisms underlying both "on-" and "off-target" effects, highlighting the strengths of different variants and the weaknesses to be addressed by future technical innovations.

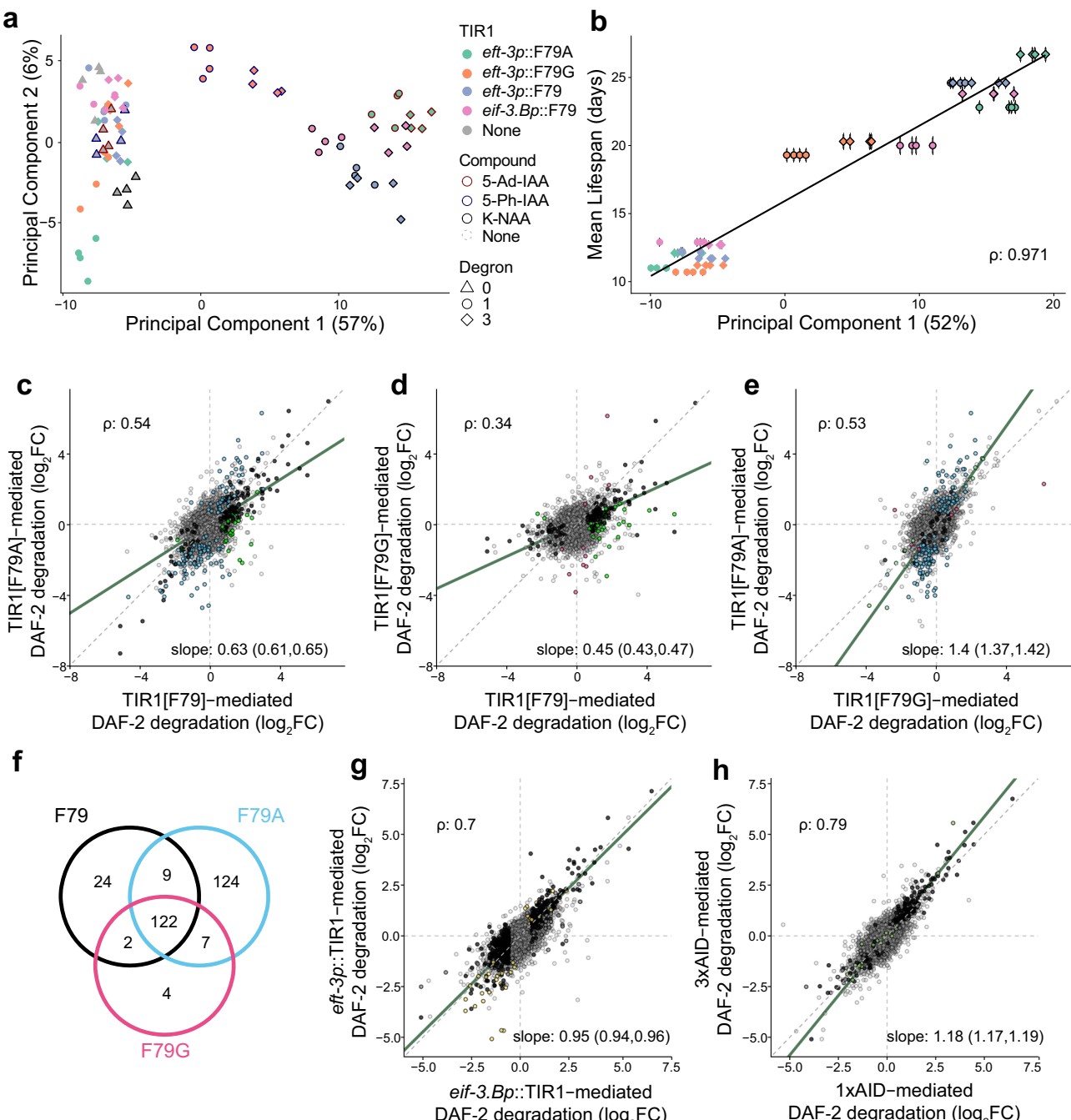

**Fig. 3 | Comparing the "on-target" effects of AID system variants.** mRNA was collected on the first day of adulthood from wild-type populations (gray) from eight strains expressing either *eft-3p*::TIR1[F79] (blue), *eif-3.Bp*::TIR1[F79] (pink), *eft-3p*::TIR1[F79A] (green) or *eft-3p*::TIR1[F79G] (orange) paired either with *daf-2*::AID (circles) or *daf-2*::3×AID (diamonds) tags, as well as a wild-type control population (gray). Populations were exposed to K-NAA (black outlines), 5-Ph-IAA (blue outlines), 5-Ad-IAA (red outlines), or no compound (no outline). **a** The gene expression of each population projected along the first two Principal Components (PC) of the data set. **b** The PC1 values of each strain compared to that strain's mean lifespan (*N* = 3604 individuals across four biological replicates), with a linear fit (line). The error bars represent ±1 SE. **c** In the presence of their respective activating compounds, the effect of DAF-2::3×AID knockdown mediated by TIR1[F79] on each gene's expression relative to the wild-type control (Statistical methods) compared to equivalent effects mediated by TIR1[F79A], **d** TIR1[F79G], and **e** TIR1[F79G] compared to TIR1[F79A]. Colored points indicate genes that are significantly differentially expressed (Adjusted Wald *p* < 0.01) in F79 (black), F79G (red), F79A (blue), and F79G and F79A but not F79 (green). **f** The overlap in significantly differentially expressed genes. **g** TIR1[F79] expressed under the *eft-3* promoter compared to the *eif-3.B* promoter, and **h** *daf-2*::1×AID compared to *daf-2*::3×AID in *eif-3.Bp*::TIR1[F79] populations. Colored points indicate genes that are significantly differentially expressed (Adjusted Wald *p* < 0.01) in *eif-3.Bp*::TIR1 (yellow) and 1×AID (green).

## A "dual-channel" AID system combines orthogonal, tissue-specific TIR1 variants

A major constraint of the AID system is that it allows the degradation of only one target at a time. However, our data (Fig. 1f) highlights an orthogonality between different TIR1 variants, which we can

leverage to create a "dual-channel" AID system to enable independent protein depletion in two different tissues. Since concentration regimes exist in which TIR1[F79] and TIR1[F79G] can be independently activated by K-NAA and 5-Ph-IAA, respectively, with no crosstalk[22,23,26] (Fig. 1f), we reasoned that both variants could be

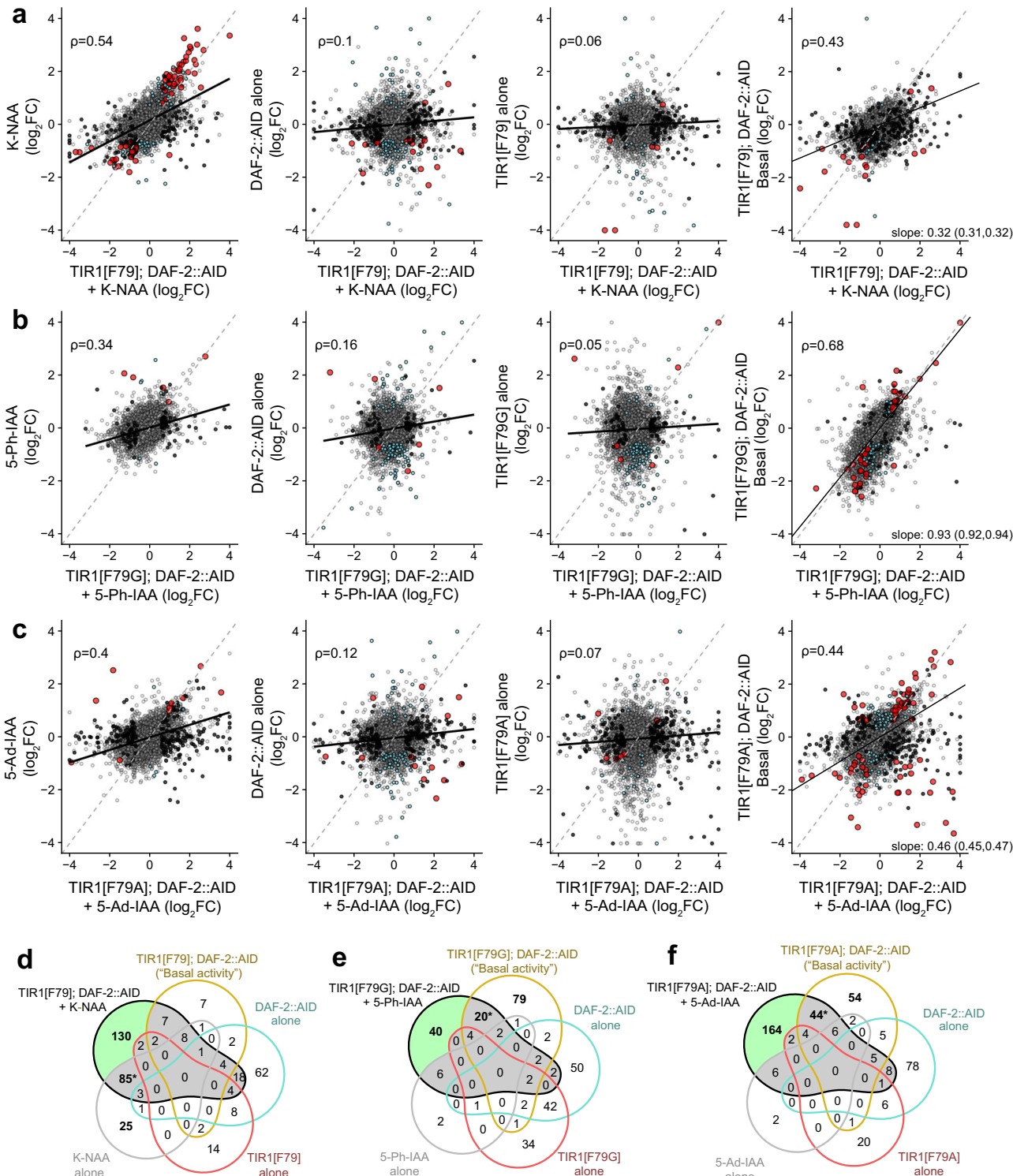

**Fig. 4 | Comparing the "off-target" effects and basal activity of AID system variants.** The effect of individual components of the AID system on gene expression, compared to the "on-target" effects of the activated TIR1; DAF-2::AID + compound triple system. **a** (left) The effect of K-NAA on each gene in wild-type populations (*y*-axis), compared to the effects of the activated triple (*x*-axis). Genes differentially expressed (Adjusted Wald *p* < 0.01) only in the activated triple (black), only by K-NAA (blue), or in both (red). (center left) The effect of the DAF-2::1xAID tag in the absence of TIR1 or K-NAA. (center right) TIR1[F79] in the absence of DAF-

2::AID or K-NAA. (right) the "basal activity" of the double TIR1[F79]; DAF-2::AID in the absence of K-NAA. **b** The same comparisons, but for combinations involving TIR1[F79G] and 5-Ph-IAA, and **c** for combinations involving TIR1[F79A] and 5-Ad-IAA. **d**–**f** Venn diagrams decomposing all effects of the activated TIR1 system into true "on-target" effects (green shaded), "off-target" effects (gray shaded) present in the activated triple, and the effects produced by AID system components singly but not observed in the activated triple.

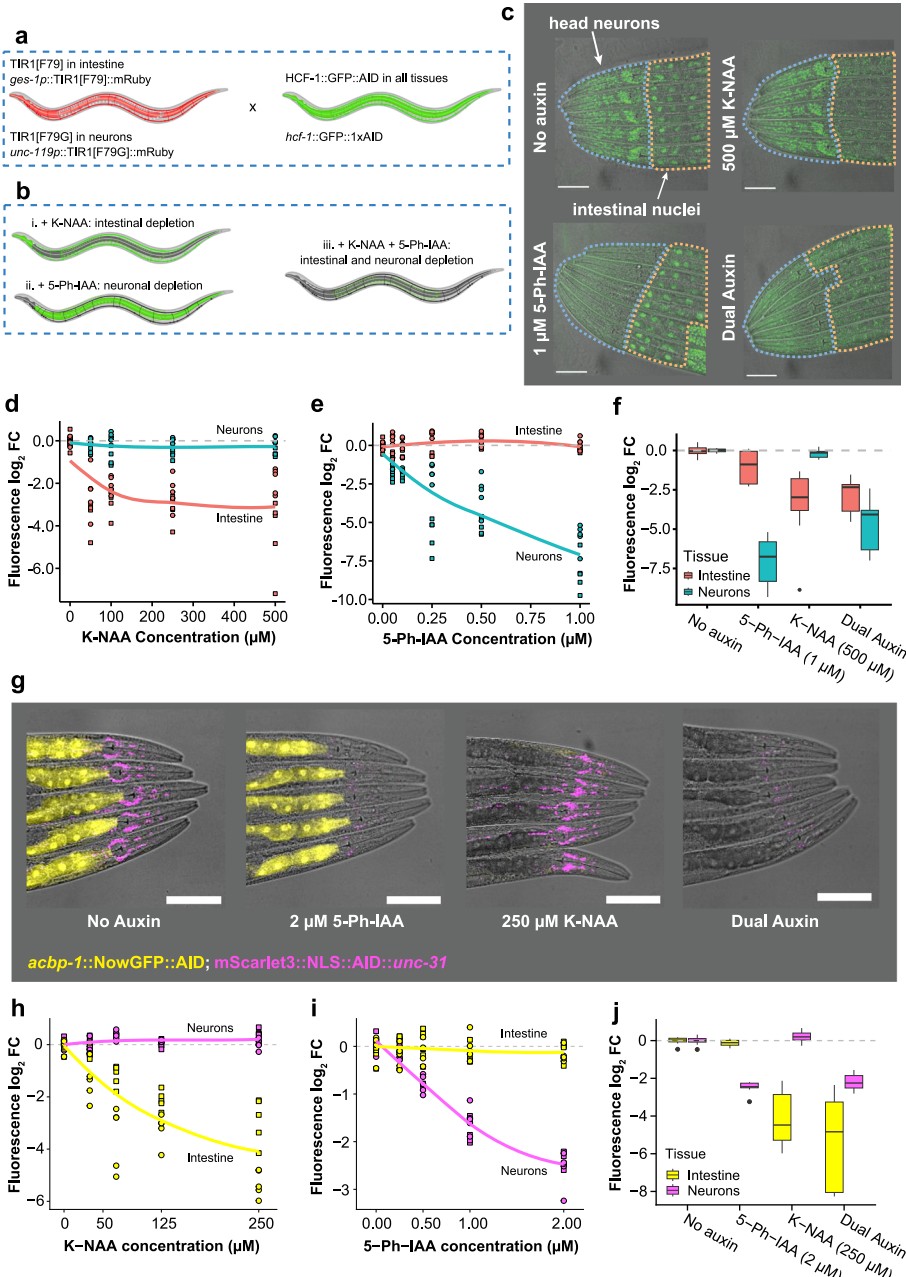

**Fig. 5 | A "dual-channel" AID system using tissue-specific TIR1 variants.** The dual-channel AID system results in tissue-specific degradation of the target of interest. **a** A strain harboring the HCF-1::GFP::degron reporter was crossed with a strain containing two tissue-specific TIR1 transgenes: *ges-1p*::TIR1 (intestine) and *unc-119p*::TIR1[F79G] (neurons). **b** Depending on the auxin analog added, HCF-1::GFP can be selectively depleted in the intestine or neurons, or simultaneously depleted in both tissues. **c** Representative image of dual-channel TIR1; HCF-1::GFP::degron worms without auxin shows localization of GFP signal in neuronal and intestinal nuclei (upper left) from two independent replicates (N = 2). Addition of 500 μM K-NAA results in degradation of the target in the intestine (upper right), addition of 1 μM 5-Ph-IAA results in degradation of the target in neurons (lower left), while addition of both compounds results in degradation in both tissues (lower right). **d** Exposure to varying concentrations of K-NAA shows a preference for the degradation of the target in the intestine. **e** Exposure to 5-Ph-IAA results in the opposite effect, with preferential degradation of the target occurring in neurons. **f** Exposure to both K-NAA and 5-Ph-IAA results in simultaneous depletion in both the intestine and neurons. **g** Representative images (from two independent replicates, N = 2) of worms expressing the double tissue-specific TIR1 strains tagged with a fluorophore and 1xAID at the ACBP-1 and UNC-31 loci (left) and exposed to 5-Ph-IAA (center left), K-NAA (center right), or both compounds (right), resulting in tissue-specific degradation. **h–j** Same comparisons as (**d–f**) but in the ACBP-1; UNC-31-tagged strain. Fluorescence log₂ FC: $\log_2$ fold change of fluorescent signal quantified using absolute photon counts normalized to the no auxin condition. In (**d–e**) and (**h–i**), points represent the signal quantified from individual worms, and the shape of the points corresponds to each independent replicate (N = 2). The data in (**f, j**) are also derived from the same set of experiments (N = 2). All scale bars = 100 μM. **a, b** were created in BioRender. Vicencio, J. (2025) https://BioRender.com/a1asink.

expressed in nonoverlapping tissues to provide independent quantitative control over a protein's degradation rate in different tissues. As a proof-of-concept, we decided to express the TIR1[F79] and TIR1[F79G] variants using intestinal and neuronal-specific promoters, respectively (Fig. 5a). The neuronal-intestinal axis is important in many biological processes in *C. elegans*, including aging[41–45], chemosensory behavior[46,47], rhythmic behavior[48], and innate immunity[49–51].

We then chose HCF-1::GFP::AID as a brightly expressed degradation target and quantitative readout of TIR1 activity (Fig. 5a). With the combination of TIR1 variants used, we hypothesized that organismal exposure to K-NAA should deplete HCF-1 specifically in the intestine. In contrast, exposure to 5-Ph-IAA should deplete HCF-1 specifically in neurons (Fig. 5b). Using fluorescence as a measure of HCF-1 abundance, we observed exactly this effect (Fig. 5c). Across replicates, we observed a dose-dependent degradation of intestinal HCF-1 in response to K-NAA (Fig. 5d) and a dose-dependent degradation of neuronal HCF-1 in response to 5-Ph-IAA (Fig. 5e). When individuals were exposed to both compounds simultaneously, we observed degradation in both neurons and the intestine (Fig. 5f). However, neuronal degradation rates appeared to be lowered slightly in the presence of both compounds compared to only 5-Ph-IAA, perhaps through the competitive occupation of TIR1[F79G] binding sites by K-NAA.

Another benefit of the dual-channel system is that it facilitates the independent degradation of two proteins expressed in non-overlapping tissues. As a demonstration, we chose ACBP-1, which is responsible for the binding and intracellular transport of acyl-CoA esters in the intestine, and UNC-31, which is responsible for dense core vesicle exocytosis in neurons. Using K-NAA and 5-Ph-IAA to degrade ACBP-1 and UNC-31, respectively, we demonstrate dose-dependent control of each protein without crosstalk (Fig. 5g–j). Based on these results, we conclude that a combination of TIR1[F79] and TIR1[F79G] with distinct tissue-specific expression patterns can be used to independently and simultaneously control two target protein levels in vivo.

### Transgene de-silencing complicates the design of a germline-soma dual-channel AID system

We then sought to generate a "dual-channel" system that would allow independent degradation of proteins in the soma and the germline. To do this, we combined the pan-somatic *eft-3p*::TIR1::*unc-54* 3'UTR variant[12] with a newly created germline-expressed *mex-5p*::TIR1[F79G]::*tbb-2* 3'UTR transgene. To visualize AID activity across all tissues, we measured RPB-2::GFP[52], the second-largest subunit of RNA polymerase II, as a bright, nuclear-localized protein visible in both somatic and germline nuclei. Unexpectedly, the combination of TIR1 expressed under somatic and germline promoters did not produce a compound-dependent, tissue-specific response (Supplementary Fig. 6a): exposure to K-NAA resulted in degradation of RPB-2::AID both in the soma (Supplementary Fig. 6b) and in the germline (Supplementary Fig 6c). Instead, we found that the K-NAA-dependent germline depletion of RPB-2::AID required both somatic *eft-3p::TIR1* and germline *mex-5p*::TIR1[F79G] transgenes, as no germline depletion was observed when either TIR1 transgene was present in isolation (Supplementary Fig. 6c). We therefore conclude that expression of the K-NAA-insensitive *mex-5p*::TIR1[F79G] transgene must somehow enable the K-NAA-sensitive pan-somatic *eft-3p*::TIR1[F79] transgene. Using confocal microscopy, we identify the mechanism responsible for this interaction: *eft-3p*::TIR1::mRuby becomes ectopically expressed in the germline in the presence of the *mex-5p*::TIR1[F79G] transgene (Supplementary Fig. 6d). Therefore, the failure of co-expressed somatic and germline TIR1 variants to provide "dual-channel" tissue-specific AID target knockdown arises from the transcriptional induction of germline *eft-3p*::TIR1 by *mex-5p*::TIR1[F79G], which then allows K-NAA to deplete RBP-2::AID both in the soma and the germline.

A major determinant of germline transgene expression is gene silencing mediated by PIWI-interacting RNAs (piRNAs)[53,54]. We therefore investigated whether the presence of the germline-licensed *mex-5p*::TIR1[F79G] might result in the de-silencing of the nearly homologous *eft-3p*::TIR1 transgene in the germline[54]. We disrupted HRDE-1 via RNAi to globally inhibit germline silencing[55], and after five generations of *hrde-1* RNAi, we began to detect *eft-3p*::TIR1 expression in the germline, even in the absence of *mex-5p*::TIR1 (Supplementary Fig. 7a). We confirmed that de-silencing was sufficient to enable K-NAA-dependent RPB-2::GFP degradation in the germline (Supplementary Fig. 7b–d). From these data, we conclude that piRNA-mediated transgene silencing is required for the pan-somatic expression pattern of *eft-3p*::TIR1 and that disruption of this silencing is sufficient to convert *eft-3p*::TIR1 into a whole-body expression pattern. We therefore conclude that a germline/soma dual-channel auxin degron system is not possible using existing TIR1 transgenes, as the required pan-somatic promoter—a promoter whose exclusion from the germline does not depend on germline silencing—has not yet been developed.

### Pan-organismal knockdown of proteins using a re-engineered TIR1

Our better understanding of *eft-3*p::TIR1 suggested a path for producing a true pan-organismal auxin-inducible degron system. Such a system would allow AID experiments that recapitulate the effect of conventional mutations and RNAi experiments, which typically involve knockdown of protein targets across all tissues. We hypothesized that such a pan-organismal AID system could be obtained by creating a new *eft-3*p::TIR1 construct engineered to circumvent piRNA activity[56]. To obtain such a construct, we sought to eliminate the germline silencing of the TIR1 transgene by including an early first intron, the endogenous Periodic $A_n/T_n$ Clusters (PATCs), and using the *gpd-2/3* intergenic sequence for co-expression[49] (Fig. 6a). In addition, we codon-optimized the entire TIR1 coding sequence and removed potential piRNA-binding sites. We chose to swap the *unc-54* 3'UTR for the β-tubulin 3'UTR (*tbb-2*), as the latter has been reported to enable expression in all germ cells[57].

Using MosTI[58], we successfully integrated a single copy of *eft-3p*::TIR1::SL2::NLS::BFP::*tbb-2* 3'UTR. However, this transgene was not expressed in the germline (Fig. 6b). Exposure to *hrde-1* RNAi for several generations again resulted in de-silencing of germline TIR1 expression (Supplementary Fig. 7e), demonstrating that our changes did not eliminate piRNA-mediated transgene silencing. Since the endogenous EFT-3 protein is expressed in the germline, we hypothesized that germline expression of TIR1 might be rescued by switching from the *tbb-2* 3'UTR to the native *eft-3* 3'UTR. We therefore performed a 3'UTR swap using CRISPR/Cas9, excising the existing *tbb-2* 3'UTR from the TIR1 transgene and replacing it with the native *eft-3* 3'UTR sequence. Happily, we find this final modification was sufficient to enable germline expression of TIR1, providing the pan-organismal TIR1 transgene: *eft-3p*::TIR1::SL2::NLS::BFP::*eft-3* 3'UTR (Fig. 6b). To confirm that this transgene exhibits TIR1 activity in both somatic tissues and the germline, we returned to the degron-tagged RPB-2::GFP as a reporter for AID activity. In contrast to the *eft-3p*::TIR1::*unc-54* 3'UTR precursor, the *eft-3p*::TIR1::*eft-3* 3'UTR transgene degrades RPB-2::AID across all tissues (Fig. 6c–e). We conclude that our investigation into piRNA-mediated transgene silencing in the AID system allowed us to generate a new pan-organismal TIR1 transgene.

## Discussion

The AID system leverages the complex interplay between small molecules and multiple transgenes to achieve precise spatiotemporal control of protein abundance in vivo. To understand and improve AID technologies, we have applied engineering approaches to isolate the biochemical factors that contribute to and limit the performance of the AID system in the multicellular animal *C. elegans*. Measuring TIR1 activity across the lifespans of 7400 adults and 122,000 dauers, we discover that TIR1 abundance and ubiquitin binding site availability interact to determine the chemical kinetics of the AID system. Using gene expression, we decompose both the "on-" and "off-" target effects

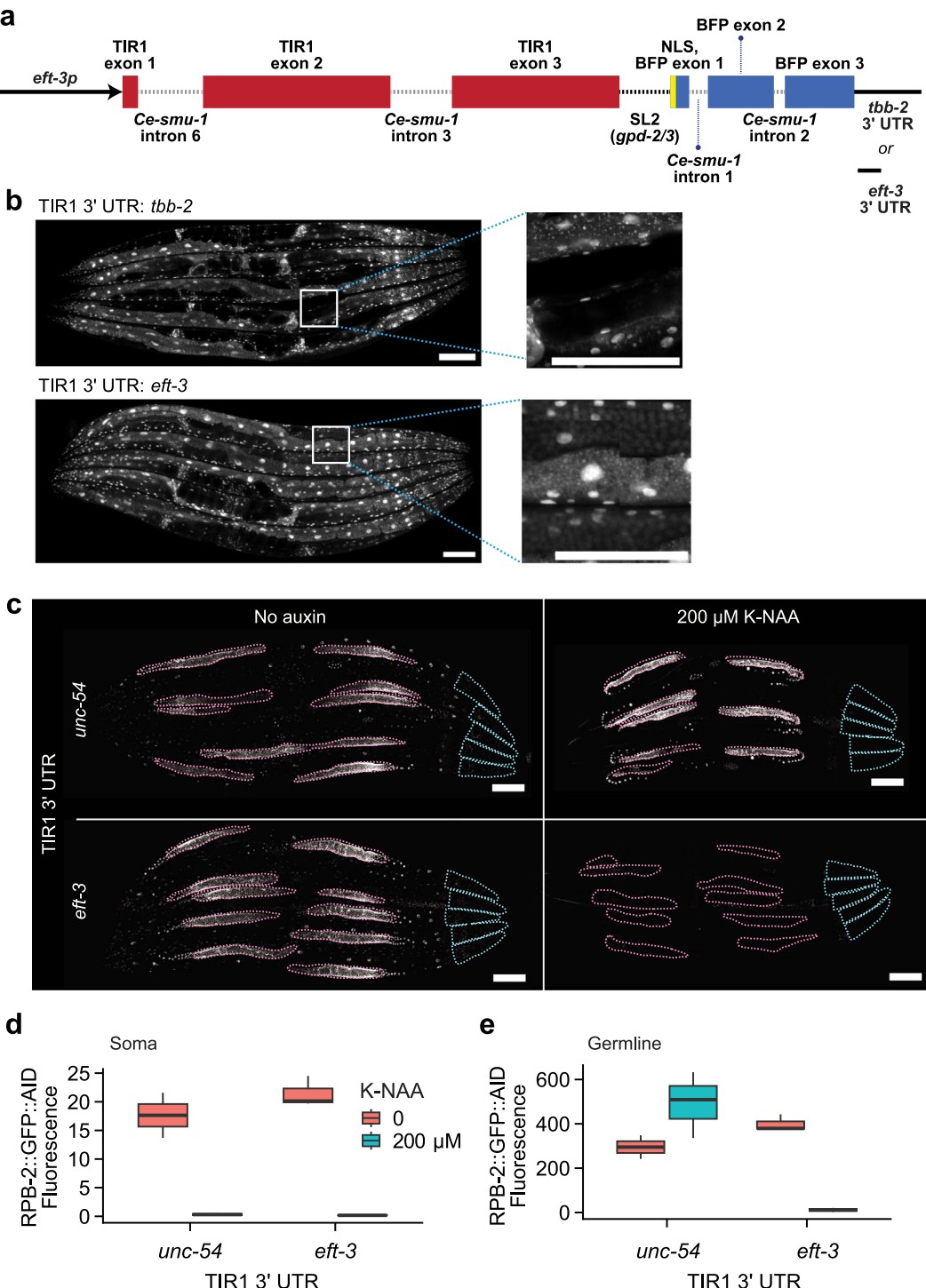

**Fig. 6 | A pan-organismal AID degradation system.** A re-engineered TIR1 trans-gene with somatic and germline expression is capable of depleting target proteins across the whole body. **a** Schematic of the redesigned TIR1 transgene consisting of the *eft-3* promoter, TIR1 with *smu-1* introns, SL2 trans-splicing sequence from the *gpd-2/3* operon, c-Myc nuclear localization signal (NLS), mTagBFP2 with *smu-1* introns, and a *tbb-2* or *eft-3* 3'UTR. The diagram is drawn to scale. **b** Confocal images of the redesigned TIR1 transgenes containing either the *tbb-2* 3'UTR or the *eft-3* 3'UTR. The insets highlight the absence or presence of TIR1::mTagBFP2 germline expression in *tbb-2* 3'UTR and *eft-3* 3'UTR animals, respectively. **c** Comparison of somatic and germline protein depletion of degron-tagged RPB-2::GFP between *eft-3p*::TIR1::mRuby::*unc-54* 3'UTR and our redesigned *eft-3p*::TIR1::mTagBFP2::*eft-3* 3'UTR transgene upon auxin exposure in five worms per condition. **d** Quantification of somatic and **e.** germline RPB-2::GFP signal of the animals shown in (**c**), in the presence or absence of auxin. Data for (**c**–**e**) are derived from five worms per condition from a single experiment. Scale bars = 100 μM.

of the AID system into its individual components and highlight that a recent improvement to the AID system, AID2, solves some problems while introducing others: eliminating the dependence on the bioactive auxin/K-NAA molecule at the cost of increasing basal degradation and lowering maximum protein degradation rates. Notably, our high-throughput phenotyping and transcriptomic analysis identify that another TIR1 variant, TIR1[F79A] (ssAID), restores the high "on-target" activity of TIR1[F79] when combined with a 3xAID degron tag at the

expense of a somewhat increased basal (auxin-independent) degradation rate. Due to its improved enzymatic activities, the TIR1[F79A] variant, which we introduce into *C. elegans*, will be more performant in most use cases compared to the AID2 TIR1[F79G].

Our quantitative understanding of the different AID variants also allows us to design and implement a "dual-channel" AID system that combines TIR1[F79] and TIR1[F79G] variants expressed in distinct tissues to effect simultaneous, independent control of target protein abundance in a tissue-specific manner. In multicellular animals, single proteins often play distinct physiological roles in different tissues[59–61], and our technique provides a new approach for interventional investigations into such phenomena by quantitatively manipulating the abundance of a single protein in two distinct tissues. Our "dual-channel" technique also enables independent manipulation of two proteins in vivo when they are expressed in nonoverlapping tissues.

Finally, we generated a re-engineered TIR1 transgene capable of robustly depleting target proteins in both the soma and the germline. In total, we demonstrate how a better quantitative understanding of the auxin-inducible degron system can be used to design technologies that support advanced experimental interventions into organismal physiology.

## Methods

### *C. elegans* strains and maintenance
*C. elegans* strains were maintained on Nematode Growth Medium (NGM) plates seeded with *E. coli* OP50 bacteria[62]. Strains were maintained at 20 °C, except in lifespan assays, where worms were kept at 25 °C. The names and genotypes of all strains generated in this study are listed in Supplementary Table 4.

### Plate preparation
Potassium 1-Naphthaleneacetate (K-NAA) and 5-Adamantyl-IAA (5-Ad-IAA) were obtained from TCI (N0006 and A3390, respectively), while 5-Ph-IAA was obtained from MedChemExpress (HY-134653). For K-NAA, a 500 mM stock solution was made by dissolving the salt in M9 buffer and stored at −20 °C. For 5-Ad-IAA and 5-Ph-IAA, 5 mM stock solutions were made by dissolving the salt in DMSO and stored at −80 °C. Before addition to molten NGM, the stock solutions were further diluted in M9 to the required final concentration. Molten NGM was kept at a temperature of 56 °C in a water bath prior to auxin supplementation and plate pouring. Plates were seeded with fresh *E. coli* OP50. There is some disagreement in the literature about whether 1 mM auxin produces TIR1-independent effects on lifespan, with some studies finding a lifespan-extending effect[44,63], others finding no effect[38,64], and others finding a lifespan-shortening effect[65]. We therefore referred to a full dose-effect study of K-NAA on lifespan and chose concentrations at or lower than 750 μM, which have no effect on lifespan[52].

### Dauer and lifespan assays
For dauer and lifespan assays, DAF-2::1×AID or DAF-2::3×AID embryos were synchronized by bleaching gravid hermaphrodites. For dauer assays, approximately 175 eggs were placed on each 55-mm plate immediately after bleaching and maintained at 20 °C. Scoring for dauer formation was performed after 48 h. Three plates were scored per condition, in three independent trials (N-2000 worms per condition). For lifespan assays, synchronized embryos were allowed to grow on NGM plates without auxin for 48 h at 25 °C (up to the late L4 stage) before being transferred to auxin plates supplemented with 5-Fluoro-2'-deoxyuridine (FUDR, Sigma-Aldrich, F0503) at a final concentration of 40 μM. Approximately 40 worms were placed per 55-mm plate, with four plates per condition. Dead and alive worms were scored approximately every other day by gently touching the worm's nose with the tip of a platinum wire. Worms that burst, climbed onto walls, or were missing were censored from the analysis. Worms surviving to day 20 of the assay were carefully transferred to fresh auxin plates. Worms were maintained at 25 °C for all lifespan assays.

### Fluorescence imaging experiments
HCF-1::GFP::AID, RPB-2::GFP::AID, and ACBP-1::NowGFP::AID; mScarlet3::NLS::AID::UNC-31 worms were synchronized by egg layoff of gravid hermaphrodites within a two-hour window. Worms were transferred to auxin plates at the L3 stage for HCF-1 experiments, at the L4 stage for ACBP-1; UNC-31 experiments, and at the D1 stage for RPB-2 experiments. Worms were exposed to different concentrations of auxin for either 3 h (ACBP-1; UNC-31) or 24 h (HCF-1 and RPB-2) prior to confocal imaging.

### DNA constructs and cloning
The donor vector for the single-copy integration of *eft-3p*::TIR1::SL2::NLS::mTagBFP2::*tbb-2* 3'UTR was constructed through Gibson cloning. The *eft-3p* and *tbb-2* 3'UTR fragments were amplified from *C. elegans* N2 genomic DNA, while the TIR1 and SL2::NLS::mTagBFP2 fragments were ordered as gBlocks from Integrated DNA Technologies (IDT). The gBlocks were codon-optimized using IDT's codon optimization tool. The codon-optimized sequences were then scanned for potential piRNA binding sites using PirScan[66], and silent mutations were introduced where necessary. Finally, introns from the *C. elegans smu-1* gene were introduced into the coding sequences of TIR1 (*Ce-smu-1* introns 6 and 3) and mTagBFP2 (*Ce-smu-1* introns 1 and 2). The *eft-3p*, gBlock1 (TIR1), gBlock2 (SL2::NLS::mTagBFP2), and *tbb-2* 3'UTR fragments were joined together using overlap extension PCR to form the whole fragment, which we designate as nTIR1. The nTIR1 fragment was cloned into the pSEM246 vector using in-house Gibson cloning reactions. The pSEM246–MCS MosTI *unc-119* target was a gift from Christian Frøkjær–Jensen (Addgene plasmid # 159821). The resulting vector was transformed into 10-beta Competent *E. coli* (High Efficiency) cells (New England Biolabs, C3019H) following the manufacturer's protocol. Selected transformants were sequence-verified using whole plasmid sequencing. The final clone harboring the TIR1 transgene and MosTI donor sequences is designated as pNES0036. All oligos and constructs are listed in Supplementary Table 5, while the complete nTIR1 sequence is supplied in Supplementary Table 6.

The degron 1× and 3× sequences were based on the AID* sequence, with the addition of an N-terminal GSGGGG linker. The degron sequences were codon-optimized using IDT's codon-optimization tool and were ordered as gBlocks. The codon-optimized 1×AID and 3×AID sequences are listed in Supplementary Table 6.

### Transgenesis
***daf-2*::1xAID and *daf-2*::3xAID**. The double-stranded repair template containing the 1×AID or 3×AID sequence was amplified from the corresponding gBlock using Phusion high-fidelity polymerase (Thermo Fisher Scientific, F530L). CRISPR/Cas9 transgenesis of the *daf-2* C-terminal target region was then performed following a standard microinjection protocol[67].

**TIR1[F79G] and TIR1[F79A]**. To generate TIR1[F79G] and TIR1[F79A], corresponding single-amino acid substitutions were introduced into the TIR1[WT] sequence in the CA1200 strain[12] using CRISPR/Cas9. The single-stranded DNA (ssDNA) repair templates were ordered as 4-nmol Ultramers from IDT. $F_1$ candidates were screened using amplification-refractory mutation system (ARMS) PCR[68]. To generate the germline-specific and neuronal-specific TIR1[F79G] transgenes *mex-5p*::TIR1[F79G]::F2A:mTagBFP2::AID*::NLS::*tbb-2* 3'UTR (*ohm52*) and *unc-119p*::TIR1[F79G]::mRuby (*ohm24*), respectively, the JDW221 and HAL227 strains containing the TIR1[WT] sequence were modified using CRISPR/Cas9 as previously described. JDW221

and HAL227 are gifts from Jordan Ward and Hannes Lans, respectively.

**eif-3.Bp::TIR1**. The *eif-3.Bp*::TIR1::linker::mCherry$^{\Delta piRNA}$::*tbb-2* 3'UTR transgene was integrated into the EG6699 strain as a single-copy transgene using MosSCI[69], resulting in the JA1880 strain. The insertion was sequence-verified through genomic DNA sequencing by performing de novo alignment of reads to contigs, followed by contig mapping to the reference genome. Contigs that contain exogenous sequences confirm a single-copy insertion of *eif-3.Bp*::TIR1::*tbb-2* 3'UTR in the *ttTi5605* Mos1 insertion site in Chromosome II.

**nTIR1**. The redesigned *eft-3p*::nTIR1::SL2::NLS::mTagBFP2::*tbb-2* 3'UTR transgene was integrated into the CFJ94 strain as a single-copy transgene in Chromosome IV using MosTI with the *unc-119* selection approach[58], resulting in the AMP228 strain. The insertion was sequence-verified via whole-amplicon sequencing. Then, the *Cbr-unc-119* rescue fragment was floxed after injection with Cre recombinase, and the resulting *unc* strains were outcrossed twice to N2 to remove the *unc-119(ed3)* allele, resulting in the AMP245 strain. The replacement of the *tbb-2* 3'UTR to the *eft-3* 3'UTR was accomplished through CRISPR/Cas9 using two guide RNAs that excised the *tbb-2* 3'UTR. The *eft-3* 3'UTR repair template was ordered as an ssDNA Ultramer from IDT. The resulting strain containing the *eft-3p*::nTIR1::SL2::NLS::mTagBFP2::*eft-3* 3'UTR (*ohm53*) transgene is designated as AMP267. We confirmed stable TIR1 germline expression in AMP267 across five generations, after several months of starvation at 20 °C, and after multiple genetic crosses with different AID-tagged proteins. In one instance, we observed reduced mTagBFP2 fluorescence and germline TIR1 activity in the *ohm53*; *rpb-2::AID* strain after several months of starvation at 20 °C, perhaps due to selection for mutations reducing the deleterious effects of basal "leaky" TIR1-mediated RPB-2::AID knockdown in the germline. Any complex, long-term interactions between TIR1 and essential proteins tagged with AID can be avoided by regularly returning to frozen strain stocks.

### RNAi experiments

CA1200 and AMP245 worms were cultured on NGM plates supplemented with 0.5 mg/mL ampicillin and 2 mM Isopropyl ß-D-1-thiogalactopyranoside (IPTG) seeded with HT115 empty vector (EV) or *hrde-1* RNAi from the Ahringer library[70]. Worms were maintained at 25 °C and transferred to fresh RNAi plates every other generation.

### mRNA sequencing

mRNA sequencing was conducted using the Smart-Seq2 technology[71], which has recently been adapted for nematode organismal transcriptomics studies[72] and has been proven to be an accurate and cost-effective method for capturing population-scale gene expression[73]. Lysis buffer was prepared according to the published Smart-Seq2 protocols, with ERCC spike-ins[74] added at a final dilution of 1:40,000. Thirty synchronized adult worms were individually picked into 120 μL of lysis buffer, with four replicates performed. The nematode suspensions were shock-frozen in liquid nitrogen and stored at −80 °C. Lysis was carried out at 65 °C for 10 min, followed by enzyme inactivation at 85 °C for 5 min. cDNA libraries were then prepared according to the Smart-Seq2 protocol and purified using in-house prepared SPRI paramagnetic beads, which mimic AMPure XP beads (Beckman Coulter), with a bead-to-sample ratio of 0.8. The size distribution of the libraries was assessed using a TapeStation 4150 (Agilent), and cDNA concentration was measured using Quant-iT (Invitrogen) on a plate reader (Tecan). Nextera sequencing libraries were prepared from the cDNA through tagmentation and subsequent PCR amplification with indexing primers, following the Nextera DNA library prep protocol (Illumina). These libraries were purified twice using in-house-prepared

SPRI paramagnetic beads at a bead-to-sample ratio of 0.9. The library size distribution was evaluated again as described above. The nematode RNAseq libraries were pooled in equal mass, and the paired-end Nextera libraries were sequenced on an Illumina NextSeq 500 using high-output 75-cycle v2.5 kits (Illumina), generating read lengths of 38 bases and yielding more than $2.0 \times 10^6$ reads per sample.

### Image acquisition

**Confocal microscopy**. Worms were anesthetized on 5% agarose pads using 5 mM of levamisole (ITW Reagents, P110801) dissolved in M9 and secured with a #1.5 coverslip. Images were acquired in fluorescence lifetime imaging (FLIM) mode using a Leica SP8 FALCON microscope equipped with a white light laser (WLL) and a 40x Plan Apochromat 1.3 NA oil immersion objective controlled by Leica Application Suite X software (version 3.5.7.23225). TIR1::BFP images for RNAi experiments were acquired using a Leica SP5 inverted microscope equipped with a 405 nm diode laser, a 40x Plan Apochromat, 1.25–0.75 NA oil immersion objective controlled by LAS AF software (version 2.6.3. 8173). AMP100 and AMP281 TIR1::BFP images and TIR1::mRuby images were acquired using a Zeiss LSM 980 microscope equipped with 405 nm and 561 nm diode lasers, and a 20× Plan Apochromat, 0.8 NA dry objective controlled by Zen blue software (version 3.3.89.008).

### Statistical methods

**Image processing and analysis**. GFP, NowGFP, and mScarlet3 images acquired in FLIM mode were thresholded for photon counts in the Leica Application Suite X software. Bi-exponential reconvolution was performed by setting the first component corresponding to autofluorescence to 0.7 ns and by fitting the second component to the fitted lifetime specific to each fluorophore. Regions of interest were drawn based on brightfield images, and fluorescence signals were quantified using a custom Python (version 3.12.3) script using the napari library. For HCF-1::GFP and mScarlet3::UNC-31 signal in the neurons, the region of interest spanned the lower half of the first bulb of the pharynx down to the end of the second bulb of the pharynx. For signal in the intestines, the region of interest covered the area demarcated by the first two pairs of anterior intestinal cells in the case of HCF-1::GFP, whereas the whole intestine was covered for ACBP-1::NowGFP. For RPB-2::GFP images, signal across the entire head region (from the tip of the head to the end of the second pharyngeal bulb) was used as a proxy for signal in somatic cells. For the signal in the germline, the region of interest covered the distal germline spanning the mitotic zone and the pachytene region before the loop. All images acquired were processed for display by cropping, rotating, and adjusting brightness and contrast using Fiji/ImageJ (version 2.14.0/1.54f). All scale bars are set to 100 μM. No additional image manipulations were performed.

**Lifespan and dauer analysis**. All data plotting and statistical analysis was performed in R (version 4.2.3). For box plots, the boxes represent the interquartile range, and the horizontal line represents the median. The whiskers extend to the minimum and maximum values that are within 1.5 x IQR of the 25th and 75th percentiles, respectively. Outliers are defined as data points beyond 1.5 x IQR and are plotted as individual points beyond the whiskers. Kaplan–Meier analysis was performed in R using the survival package. Accelerated failure time (AFT) regression models were fit using Buckley–James regression with the rms package. Hill coefficients for dauer and lifespan data were fit to AFT regression coefficients using the formula $E(D) = E_0 + (E_f - E_0)/(1 + D^{-n}\ln(IDM))$ using braidrm, with $E_0$ fixed and $E_f$, $n$, and IDM estimated.

**Transcriptomics pre-processing**. RNA-seq reads were aligned to the *C. elegans* Wormbase reference genome (release WS265) using STAR version 2.6.0c, with modifications to include ERCC spike-in sequences. Gene counts were quantified with featureCounts version 2.0.0. The

count matrix was filtered using a detection threshold that required at least 5 counts in at least half of the samples. Genome coverage of the reads was calculated using BEDTools. The sequence read archive (SRA) for our transcriptomic data can be found under BioProject number PRJNA1167633.

**Transcriptomics analysis: regression analysis of AID component effects.** To estimate the "on-target" effects of TIR1-mediated DAF-2::AID in the presence of auxin analog (Fig. 3c–h), we used the multiple regression model: $\log y_i = \beta^{Auxin} X_i + \beta^{TIR1} W_i + \beta^{Cross} X_i W_i + \varepsilon_i$, with $y_i$ as the expression of a gene in an individual $i$, $X_i$ as the categorical variable coding for the auxin analog to which individual $i$ was exposed (K-NAA, 5-Ph-IAA, 5-Ad-IAA, or none), and $W_i$ coding for the genotype of individual $i$: *daf-2*::1xAID;TIR1[F79], *daf-2*::1xAID;TIR1[F79G], *daf-2*::1xAID;-TIR1[F79A], and "no TIR1" for Fig. 3 c–f and *eft-3p::TIR1[F79]; daf-2::1xAID, eif-3.Bp::TIR1[F79]; daf-2::1xAID, eft-3p::TIR1[F79]; daf-2::1xAID, eif-3.Bp::TIR1[F79]; daf-2::3xAID* and "no TIR1" for Fig. 3 g–h. Significant "on-target" effects of each TIR1 variant were identified as contrasts of $\beta^{Cross}$ significantly different from zero, e.g., the "on-target" effects of *daf-2*::AID;TIR1[F79] are $(\beta^{Cross} X_{K-NAA} W_{TIR1[F79]}) - (\beta^{Cross} X_{none} W_{noTIR1})$. Significant differences between TIR1 variants were calculated in the same way, e.g., the difference between *daf-2*::AID;TIR1[F79] and *daf-2*::AID;-TIR1[F79G] is $(\beta^{Cross} X_{K-NAA} W_{TIR1[F79]}) - (\beta^{Cross} X_{5-Ph-IAA} W_{TIR1[F79G]})$. The "overall effect" of the AID system in the x-axes of Fig. 4a–c is calculated as the sum of contrasts across both $\beta^{Auxin}$ and $\beta^{TIR1}$, e.g., the effect of TIR1[F79] + K-NAA is $(\beta^{Auxin} X_{none} + \beta^{TIR1} W_{noTIR1} + \beta^{Cross} X_{none} W_{noTIR1}) - (\beta^{Auxin} X_{K-NAA} + \beta^{TIR1} W_{TIR1[F79]} + \beta^{Cross} X_{K-NAA} W_{TIR1[F79]})$.

**Transcriptomics analysis: significance testing.** All regression models were estimated using DESeq2, with significant coefficients identified using the two-tailed Wald test adjusted for multiple hypotheses using the Benjamini–Hochberg procedure at a $p$-value threshold of 0.0001. To focus on biologically significant changes with meaningful effect sizes, after applying the $p$-value threshold, we excluded effects with small magnitudes—less than 50% increases or decreases, i.e., effects between 0.66 and 1.5-fold—from our lists of differentially expressed genes.

### Reporting summary

Further information on research design is available in the Nature Portfolio Reporting Summary linked to this article.

## Data availability

Strains and reagents are available upon request. The raw lifespan, dauer assay, and fluorescence data generated in this study are provided in the Source Data file. The sequence read archive (SRA) for our transcriptomic data can be found under BioProject number PRJNA1167633. Source data are provided with this paper.

## Code availability

All analysis code is provided at https://github.com/nstroustrup/AID-engineering/.

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

## Acknowledgements

We thank Joy Alcedo (Wayne State University), Julián Cerón (IDIBELL), Karen Thijssen, and Hannes Lans (Erasmus University Medical Center), Rhys McDonough (University of Cambridge), and Matt Ragle and Jordan Ward (University of California, Santa Cruz) for nematode strains; and Christian Frøkjær–Jensen and Sonia El-Mouridi (KAUST) for providing MosTI strains and reagents. We also thank Yekaterina Makeyeva and Masaki Shirayama (University of Massachusetts Chan Medical School) for fruitful scientific discussions about TIR1 germline expression. We are grateful to the CRG Core Technologies Program for their support and assistance in this work, including the CRG Advanced Light Microscopy Unit (ALMU). This work was technically supported by the EMBL Genomics Core facility. Research for this publication has been partially conducted at the Barcelona Collaboratorium for Modeling and Predictive Biology. Some strains were provided by the CGC, which is funded by the NIH Office of Research Infrastructure Programs (P40 OD010440). We acknowledge the support of the Spanish Ministry of Science and Innovation through the Centro de Excelencia Severo Ochoa (CEX2020-001049-S, MCIN/AEI/10.13039/501100011033), the Generalitat de Catalunya through the CERCA program, and the EMBL partnership. We acknowledge support from the Spanish Ministry of Economy, Industry and Competitiveness, project BFU2017-88615-P, co-funded by the European Regional Development Fund (ERDF, EU), and the Spanish Ministry of Science and Innovation (MICIN/AEI/10.13039/501100011033), project PID2020-115189GB-I00, and co-funded by ERDF, EU. These results are part of a project that has received funding from the European Research Council (ERC) under the European Union's Horizon 2020 research and innovation program (Grant agreement No. 852201). J.A. acknowledges funding from Wellcome (217170) and the Medical Research Council (MR/S021620/1).

## Author contributions

N.S. and J.V. conceived the project and designed the research. J.V., D.C., and J.A. generated *C. elegans* strains. Dauer and lifespan assays were performed by D.C. and L.S. The development of the dual-channel AID system and nTIR1 was performed by J.V. and N.S. Transcriptomics was performed by M.E., J.V., and D.C. Confocal imaging and image analysis were performed by J.V. and L.S. N.S., J.V., D.C., L.S., and M.E. performed data analysis. N.S. and J.V. wrote the manuscript with input from D.C., M.E., L.S., and J.A.

## Competing interests

All authors declare no competing interests.
