## [Transparent Peer Review file · Nature Communications]

Engineering the auxin-inducible degron system for tunable *in vivo* control of organismal physiology

Corresponding Author: Dr Nicholas Stroustrup

Version 0:

Reviewer comments:

Reviewer #1

(Remarks to the Author)

The here presented revised manuscript by Vincencio et al characterises and compares TIR1 dependent degron systems for controlling protein stability in *C. elegans*. The manuscript has significantly improved since the previous version and I am really impressed at how much the authors added, which is partially at lot more than requested.

They have added data on lifespan, confirmed the stability of the auxin like chemicals, presented and tested a new application for the here presented system, a dual-channel' that allows in parallel visualisation and separate degradation of two target proteins and finally as requested by me in the previous round, have gone above and beyond to fully characterised the 'leaky genes' of individual TIR1 variants.

I find all the presented data solid, well presented and described and most importantly very interesting and exciting.

Being a plant auxin biologist, I cannot fully comment on the lifespan experiments raised by reviewer 2, but I find the authors argument and data convincing.

Therefore I recommend to accept this manuscript and look forward to see it published

(Remarks on code availability)

I do not have the expertise to evaluate codes

Reviewer #3

(Remarks to the Author)

The revised manuscript is very much improved. I applaud the authors' efforts to include new data and analyses that have substantially improved the impact and clarity of the work. The authors provide extensive data on protein depletion kinetics by decomposing the AID system. They clearly discuss the lessons learnt and guide the reader on the advantages and disadvantages of each iteration of the AID system.

At this point, I only have some minor comments/suggestions.

1. Although the conclusions of this work are likely broadly applicable to other systems, the Abstract should mention that the work has been done in *C. elegans*, as this is the experimental model for all the experiments.

2. Figure 4 is dense and hard to follow. Consider restructuring.

3. To help the reader follow closely the data, the authors could cite more often in Results the corresponding figure panels. For example, Fig. 2e is not cited in the Results.

(Remarks on code availability)

Response to Reviewers

Reviewer #1 (Remarks to the Author)

The here presented revised manuscript by Vincencio et al characterises and compares TIR1 dependent degron systems for controlling protein stability in *C. elegans*. The manuscript has significantly improved since the previous version and I am really impressed at how much the authors added, which is partially at lot more than requested.

They have added data on lifespan, confirmed the stability of the auxin like chemicals, presented and tested a new application for the here presented system, a dual-channel' that allows in parallel visualisation and separate degradation of two target proteins and finally as requested by me in the previous round, have gone above and beyond to fully characterised the 'leaky genes' of individual TIR1 variants.

I find all the presented data solid, well presented and described and most importantly very interesting and exciting.

Being a plant auxin biologist, I cannot fully comment on the lifespan experiments raised by reviewer 2, but I find the authors argument and data convincing.

Therefore I recommend to accept this manuscript and look forward to see it published

(Remarks on code availability)

I do not have the expertise to evaluate codes

We thank the reviewer for their feedback.

Reviewer #3 (Remarks to the Author)

The revised manuscript is very much improved. I applaud the authors' efforts to include new data and analyses that have substantially improved the impact and clarity of the work. The authors provide extensive data on protein depletion kinetics by decomposing the AID system. They clearly discuss the lessons learnt and guide the reader on the advantages and disadvantages of each iteration of the AID system.

At this point, I only have some minor comments/suggestions.

We thank the reviewer for their positive feedback and suggestions.

1. Although the conclusions of this work are likely broadly applicable to other systems, the Abstract should mention that the work has been done in *C. elegans*, as this is the experimental model for all the experiments.

We have reworded the abstract to match formatting limitations and for clarity. We now included the model organism *Caenorhabditis elegans* in the abstract.

2. Figure 4 is dense and hard to follow. Consider restructuring.

We have restructured Figure 4 to make better use of the the available space.

3. To help the reader follow closely the data, the authors could cite more often in Results the corresponding figure panels. For example, Fig. 2e is not cited in the Results.

We now cite all figure panels, and include more frequent references to figures in the main text.

(Remarks on code availability)